# Extracting the Resistive Current Component from a Surge Arrester’s Leakage Current without Voltage Reference

**DOI:** 10.3390/s21041257

**Published:** 2021-02-10

**Authors:** Vid Vončina, Jože Pihler, Miro Milanovič

**Affiliations:** 1Izoelektro d. o. o., 2341 Limbuš, Slovenia; 2Faculty of Electrical Engineering and Computer Science, University of Maribor, 2000 Maribor, Slovenia; joze.pihler@um.si (J.P.); miro.milanovic@um.si (M.M.)

**Keywords:** IoT, leakage current, Metal Oxide Surge Arrester, remote monitoring, resistive component of leakage current

## Abstract

This article presents the development of the theoretical background and the design of an electronic device for monitoring the condition of a gapless Metal Oxide Surge Arrester (MOSA). The device is intended to be used online. Because of the inaccessibility and possible remote location of most surge arresters, it is equipped with a communication system, allowing for the device to convey the measurement of the surge arrester characteristics under any conditions. It is possible to determine the condition of the MOSA by gathering measurements of the surge arrester’s resistive component of leakage current. The leakage current information is sent via data transfer unit to a server and, after interpretation, will be forwarded to the authorised personnel through the surge arrester control centre.

## 1. Introduction

Crucial components in transmission and distribution networks are protected against overvoltage by MOSA (Metal Oxide Surge Arresters). For this reason, it is always of high importance to continuously monitor their condition. According to recent trends, monitoring is the most reliable way to decrease network losses and increase the reliability of equipment. Zinc oxide varistors (ZnO varistors) are the main components of MOSA, which are semiconductors that have a non-linear characteristic. This means that the voltage–current relationship is not linear, and the resistance of the element varies in regard to the applied voltage. This characteristic allows for the MOSA to perform its function.

There are several proposed methods that explain the conduction mechanism of the ZnO varistors. All of these methods take the time and exposure of the MOSA to the overvoltage into consideration. The conduction mechanism characteristics degrade to the point, where the MOSA does not perform its function adequately and, eventually, does not perform its function at all.

The frequent and high overvoltages, moisture in the housing of the MOSA, overheating, etc. are important factors that reduce the reliability of the MOSA. Overvoltage phenomena can be classified in to two groups, as:atmospheric discharges, andconsequences of switch devices’ manipulations.

Both of these have a destructive effect on the elements of the power system and they reduce the reliability of the power system’s operation. Therefore, the role of MOSA is twofold: namely, to start operating immediately when an overvoltage occurs and, secondly, to minimise the losses within the network due to excessive leakage current at continuous operation of electrical devices.

Several different methods have been identified to help determine the MOSA’s condition. These can be grouped into online and offline methods. Online testing methods are very commonly used. They tend to be simple to use and are more affordable. They range from fault indicators, disconnectors, and surge counters to more sophisticated methods, such as the failure analysis of metal oxide arresters under the harmonic distortion method [1], leakage current measurements [2,3,4,5,6,7,8], temperature measurements [9,10,11,12], and electro-magnetic field measurements [13], using Genetic Algorithms [14,15]. Another example of such a method is also the Capacitive Current Compensation Method (CCCM), which is explained in [5,16], which compensates the capacitive component to extract the resistive component *I*_r_, which is the main indicator of the state of the surge arrester. With the help of finite element analysis [8], leakage currents can be determined under different conditions. A current decomposition-based method [17] can also be used. Here, the total leakage current is decomposed to study the impact of different parameters on the decomposition accuracy. Additionally, many methods have been introduced for measuring the resistive component of the total leakage current [18,19]. However, researchers in the area have also paid significant attention to evaluating the surge arrester’s condition to reduce network losses. An arrester is considered to be faulty when the resistive current value exceeds 0.6 mA, according to the IEC 60099-5:2013, 2013 [20]. If the arrester’s current exceeds this value and is not replaced, this produces additional power losses, financial losses, and a greater carbon footprint [21], as shown in Table 1.

On the basis of the above-mentioned measuring methods, many different innovative devices for monitoring the state of surge arresters have been made available to the market. The devices presented in patents [22,23,24,25] describe a measurement system that determines the state of the MOSA based on the entire leakage current of the MOSA. Other devices [26,27] employ an internal temperature measuring device to predict the state of the MOSA. This greatly differs from the proposed method in this article, which concentrates on the procedure of extracting the resistive current precisely from the total leakage current and using it as a reference for determining the state of the MOSA. This is done without the use of voltage as a reference, which simplifies the measurement greatly and reduces costs.

The offline methods are performed in laboratories and they are more expensive, due purely to the manipulation involved in dismantling and transporting of the MOSA to an adequate facility where all of the tests can be performed or supplied with an independent voltage source for testing. Such a method is the Point On Wave Method (POWM) that is described in [28], which does not require the measurement of voltage.

This paper deals with the development of the principle and electronic device capable of a real-time arrester monitoring operation. Section 2 describes the physical background for the arrester condition monitoring principle, based on the leakage current measurement and analogue and digital acquisition of the collected measurement data. The measurement device is designed and described in Section 3 based on the theoretical conclusion. It consists of three main parts: the analogue and digital signal acquisition and communication units. Section 4 deals with experimental verification of the algorithm and electronic device.

## 2. Physical Background of the Arrester Monitoring Principle

The functionality of the arrester monitoring system is based on the physical background of the leakage current study. From the measured leakage current (*I_t_*), it is necessary to extract the resistive component (*I_r_*). During long-time operation, the arresters encounter thermal stresses, due to the power losses that are caused by the leakage current. The power dissipation can be calculated, as follows:(1)$$P_{d}=\frac{W}{T}=\frac{1}{T}\int_{0}^{T}U\left(\omega t\right)I_{t}\left(\omega t\right)d\omega t$$
where *P_d_* represents the average power-dissipation, *W* represents the energy, *T* is the period of voltage *U*(ω*t*) that causes the current *I_t_*(ω*t*), and ω represents the angular frequency. The current through a brand-new arrester is relatively small (the order of a few 10 µA), but, due to the high voltage, the dissipation becomes significant as the arrester deteriorates. It is also known that the voltage is contaminated by high order harmonics (the voltage in the observed power system is contaminated with a third harmonic *U*_3_ = 0.1% of *U*_1_, fifth harmonic *U*_5_ = 0.3% of *U*_1_ and e.c.t.), mainly with the third and fifth harmonic components, so, the voltage and current in (1) can be described as the sum of higher harmonic components, as follows:(2)$$U\left(\omega t\right)=\sum_{n=1}^{\infty }{\hat{U}}_{n}\sin\left(n\omega t\right)$$
(3)$$I_{t}\left(\omega t\right)=\sum_{n=1}^{\infty }{\hat{I}}_{t,n}\sin\left(n\omega t\right)$$
where ${\hat{U}}_{n}\;\mathrm{and}\;{\hat{I}}_{tn}$ represent the magnitudes of the *n*th voltage and current harmonic components, respectively. After the substitution of (2) and (3) into (1), and, after a short manipulation, the equation is as follows:(4)$$\begin{array}{ll} P_{d} & =\frac{{\hat{U}}_{1}{\hat{I}}_{t,1}}{2}\cos\Phi +3\frac{{\hat{U}}_{3}{\hat{I}}_{t,3}}{2}\cos3\Phi +5\frac{{\hat{U}}_{5}{\hat{I}}_{t,5}}{2}\cos5\Phi +\cdots n\frac{{\hat{U}}_{n}{\hat{I}}_{t,n}}{2}\cos\left(n\Phi \right)\\ & =U_{1}I_{t,1}\cos\Phi +3U_{3}I_{t,3}\cos3\Phi +5U_{5}I_{t,5}\cos5\Phi +\cdots nU_{n}I_{t,n}\cos\left(n\Phi \right), \end{array}$$
where $U_{n}={\hat{U}}_{n}/\sqrt{2}$ and $I_{t,n}={\hat{I}}_{t,n}/\sqrt{2};n=1,3,5,\ldots$ represent the root mean square (rms) values of the voltages and currents, respectively, and Φ is the phase delay between current and voltage. The current resistive component represents the measure of the arrester’s quality, when *I_r,rms_* exceeds the value that is defined by the Standard [29] $\left(I_{r,rms}>600/\sqrt{2}\mu A\right)$, the arrester is in bad condition, and it should be replaced by a new one. A few different MOSA (ten samples) with a nominal voltage of 36 [kV]_rms_ were randomly chosen for measurements, in order to design, in advance, so-called “static characteristics”. By presuming that the arresters are made from the same materials under the same production conditions and connected to the same supply voltage *U*, it is expected that they have statistically similar static characteristics as:(5)$${\hat{I}}_{r}={\hat{I}}_{r}\left(\varphi \right)$$

The measurement principle is based on the classical off-line measurement procedure that is suggested in [28]. The magnitude of the resistive current component (${\hat{I}}_{r}$) is extracted from the oscillograms at the instant when the voltage is at its peak (*dU*/*dt* = 0), as shown in Figure 1a. Figure 1b shows the phasor diagram at ω*t* = φ of leakage current *I_t_*, and its resistive and capacitive components *I_r_* and *I_c_*, respectively. The harmonic distortion that was obtained as a projection of the phasor *I_t_* to the real (*Re*) axis causes that distortion of the resistive current is acceptable, and the magnitude ${\hat{I}}_{r}$ is accurate enough for further analyses. However, at the same time when ${\hat{I}}_{r}$ is extracted from the diagram, the grey shaded areas that are indicated in Figure 1a,c as *I_t_avg_* are also extracted, and it depends on φ and ${\hat{I}}_{r}$, as follows:(6)$$I_{t\_avg}=I_{t\_avg}\left(\varphi ,\;{\hat{I}}_{r}\right)$$
and it represents an average value of the current *I_t_*, in the chosen interval (for example π/3≤ωt≤π), as follows:(7)$$I_{t\_avg}=\frac{3}{2\pi }\int_{\pi /3}^{\pi }I_{t}\left(\omega t\right)\;d\omega t$$

In order to extract the resistive current component $\left({\hat{I}}_{r}\right)$ from the leakage current (*I_t_*) the measurement is performed at a few different operating points (different supply voltages *U*). The measurement set-up (the following equipment was used in the set-up: A Tr1-autotransformer up to 0.4 kV, a Tr2-transformer for galvanic isolation 1/N1 = 0.4/7 kV, a high voltage transformer, 1/N2 = 0.1/110 [kV], an oscilloscope RIGOL DS4034, multimeter Fluke 189. Arrester type SNO_U = 36 kV class DH producer IZOELEKTRO) is organised, as presented in Figure 2. Figure 3a–f show the measurement results obtained from CSV data (from an oscilloscope), of a randomly chosen arrester (Arrester type SNO_U = 36 kV class DH producer IZOELEKTRO, 2341 Limbuš, Slovenia). The magnitude of the resistive current component (${\hat{I}}_{r}$) is extracted from the oscillograms at the instant when the voltage is at its peak (*dU*/*dt* = 0) at six operating points. Further, the measurements were performed on 10 arresters, in order to have a representative result. Table 2 shows the obtained measurement results. For every sample, the exact value of the phase angles (φ) and the magnitudes of ${\hat{I}}_{r}$, are indicated for every operating point.

### 2.1. Extraction of the Resistive Current Magnitudes from the Arrester’s Leakage Current

From Table 2, it is evident that there are a few measurement results showing every arrester pair φ and ${\hat{I}}_{r}$. From these values, it is convenient to calculate the average values, which usually represent the most probable values. Accordingly, the statistical parameters as average values for phase delay (φ) are:(8)$$\varphi =\frac{1}{N}\sum_{k=1}^{N}\varphi_{k}$$
the average values of resistive current magnitudes (${\hat{I}}_{r}$),
(9)$${\hat{I}}_{r}=\frac{1}{N}\sum_{k=1}^{N}{\hat{I}}_{r,k}$$
the other statistical parameters, such as the relative Standard Deviation of phase delay (σ_φ_) and of current magnitudes ($\sigma_{{\hat{I}}_{r}}$)
(10)$$\sigma_{\varphi }=\frac{1}{\varphi }\sqrt{\frac{\sum_{k}^{N}{\left(\varphi_{k}-\varphi \right)}^{2}}{N-1}},$$
(11)$$\sigma_{\hat{I}r}=\frac{100}{{\hat{I}}_{r}}\sqrt{\frac{\sum_{k}^{N}{\left({\hat{I}}_{r,k}-{\hat{I}}_{r}\right)}^{2}}{N-1}}$$
are calculated for the same operation points, and they are indicated in Table 2. The numbers extracted from the waveforms in Figure 3a–f are indicated in Table 2 as *sample* 3 results.

### 2.2. Description of the Measurement Principle for the Average Value of Leakage Current at Certain Intervals

Such a measurement of leakage current can only be performed off-line, as can be concluded from the above description. This method is not appropriate for real-time measurement, with the requirements that the voltage *U* should not be measured because of safety reasons of the whole system. When only leakage current is measured, there is no information about phase delay. However, it can be noticed, as shown in Figure 4, that the magnitude $\hat{I}r$ corresponds with exactly one average value of *I_t_* evaluated in the interval π/3≤ωt≤π. This interval can be chosen arbitrarily; in this case, it was chosen that the third harmonic component has no influence on average value, so the average value *I_t_avg_*, for the real-time measurement, can be evaluated, as follows:(12)$$I_{t\_avg}=\frac{3}{2\pi }\int_{\pi /3}^{\pi }I_{t}\left(\omega t\right)\;d\omega t=\frac{1}{n-k}\sum_{m=k}^{m=k+n}I_{t}{}_{\left(m\right)}$$
where *k* is the starting point index and *n* is the end-point index of current instants (Figure 4). The average values (*I_t_avg_*) were calculated for different arresters at different operating points, and they are collected in Table 3. The statistical parameters, the average phase delay φ, are calculated by (8), and the mean value *I_t,mn_*, which corresponds with the phase delay, is calculated, as follows:(13)$$I_{t,mn}=\frac{1}{N}\sum_{k=1}^{N}I_{t\_avg}{}_{,k}$$
and shown in Table 3 (grey shaded columns). Additionally, the relative Standard Deviations for the average of *I_t,mn_* at certain interval (σ*_It,mn_*), are calculated at the same operation points as:(14)$$\sigma_{It,mn}=\frac{100}{I_{t,mn}}\sqrt{\frac{\sum_{k}^{N}{\left(I_{t\_avg,k}-I_{t,mn}\right)}^{2}}{N-1}}$$

And they are shown in Table 3. Based on the results presented in Table 2 and Table 3, the so-called “static” characteristics are drawn to obtain analytical expressions. The obtained curves that are shown in Figure 5 can be approximated by sectional linear function. Because $I_{t,mn}(\varphi )$ is the consequence of ${\hat{I}}_{r}(\varphi )$, the intersection of the linear function appears at the same argument, when $\varphi =\varphi_{i}$ (calculating intersection of (15) and (17) gives φ_*i*_ = 164.2°, *I*_*t*,*mn*,*i*_ = 363 *μA*). Hence, the trend lines, represented by the linear function, are:when $60^{{}^{\circ}}<\varphi \leq \varphi_{i}$;
(15)$${\hat{I}}_{r}\left(\varphi \right)=-95.7\varphi +15,793$$
(16)$$I_{t,mn}\left(\varphi \right)=-38.1\varphi +6615.6$$when $\varphi_{i}<\varphi \leq 180^{{}^{\circ}}$;
(17)$${\hat{I}}_{r}\left(\varphi \right)=-9.03\varphi +1593.7$$
(18)$$I_{t,mn}\left(\varphi \right)=-7.56\varphi +1603.9$$

Both of the characteristics, for certain arrester types, must be measured in advance, and then prepared for further customer application (for example, these must be available in the arrester’s data sheet).

### 2.3. Verification of the Proposed Algorithm

The algorithm was verified while considering (15) to (18) after measuring of the unknown arrester, but it was extracted from a set of the same type. The arrester leakage current was sampled in the range φ∈(60, 180)[°] and the value of *I_t_avg_* was obtained. Figure 6 shows the proposed algorithm as a Unified Modeling Language (UML) diagram.

Table 4 shows the obtained results. The current, indicated by *I_r_cal._*, was obtained by using the formulas according to the block diagrams that are shown in Figure 6.

The results presented in Table 4 were recalculated according to the data chosen randomly from Table 2 and Table 4. In the first column are data from Table 3, the second and third columns present the recalculated phase delay and magnitudes of unknown resistive current ${\hat{I}}_{r\_cal.}$, and in the fourth column are the collected “oscilloscope” data from Table 2. The relative error was also calculated, and it was always inside the interval $\varepsilon \in \left(-2\sigma_{\hat{I}r},2\sigma_{\hat{I}r}\right)$ defined by the relative Standard Deviation (Table 2). The predicted limit of resistive current component is shown in the last row in Table 4, which appears at *I_t,mn_* = 570 μA with the phase delay of φ = 159°. Accordingly, every measured *I_t,mn_* above this number indicates that the arrester should be replaced with a new one.

## 3. Surge Arrester Monitoring Device (SAMD)

Analogue input devices, such as an Analogue-to-Digital Converter (ADC) in conjunction with a separate operational amplifier (op-amp), were used to convert a time-varying analogue signal (*I_t_*) into digital representations for further analyses in the microcontroller or a Personal Computer (PC). The proposed circuit architecture deals with the smart sensor structures that are described in more details in [30,31,32,33].

Figure 7 shows the block scheme of the proposed electronic circuit. An ADC is used with an appropriate acquisition circuit in order to obtain leakage current (*I_t_*) values (samples). As a part of the conversion, the signal acquisition circuit is designed as an instrumentation amplifier. The key characteristics are high input impedance, high common-mode rejection, low output offset, and low output impedance. After digitalisation of the leakage current (*I_t_*), it is necessary to extract the resistive current component ${\hat{I}}_{r}$ using the algorithm that is based on the UML diagram shown in Figure 6. The resistive component can be calculated in the micro-controller, or digitalised leakage current can be sent through communication channels to the server, where the ${\hat{I}}_{r}$ can be calculated offline, but in “soft” real time.

Based on the above description, it can be summarised that the measurement device must fulfil the following requirements:galvanic isolation of the measured signal (safety reasons);acquire and prepare the analogue signal for digital conversion;collection of current samples in the range $\varphi \in (\varphi_{\min},\;\varphi_{\max})$; and,prepare information for wireless transfer to an off-line server.


### 3.1. Electronic Circuits

The analogue acquisition unit was organised as a transimpedance op-amp with appropriate gain and common mode rejection ratio. Such a process requires precise and stable measurement of “small” alternative current (measuring range $I_{t}\in (-3\;\mathrm{mA},+3\;\mathrm{mA})$) when the op-amp is single voltage supplied. The current *I_t_* was captured by using a current transformer with the appropriate ratio (1:300). Such a ratio was chosen as a compromise between the sensitivity to measurement of leakage current and the rejection of the currents’ strokes of a few hundreds of an Ampere ($I_{t}:I_{t,stroke}=1:10^{7}$). This was solved by using an appropriate protection of the op-amp inputs. The battery voltage measurement and temperature measurement circuits were also integrated into the device.

#### 3.1.1. Protection Circuit

The whole measurement circuit was protected over the current’s strokes, and some high frequency interferences were rejected (Common Mode (CM) and Differential Mode (DM) interferences). Figure 8 shows the over-current protection that was performed using two Schottky and four Zener diodes. All of the diodes were chosen in order to keep the voltage difference smaller than the supply voltage (*V*_2_ − *V*_1_ < *U_p_*) ($U_{p}=3\;V$).

The high frequency interferences (noises) were reduced by using an LC filter, which contained a Common Mode (CM) choke coil (L_cm_), two line-bypass capacitors (C_cm_) for suppressing the *CM* noise signals and an across-the-line capacitor (C_dm_) for suppressing the Differential Mode (DM) noise signals.

#### 3.1.2. Signal Acquisition Circuit

Hereinafter referred to as the circuit in Figure 8, the current transformer secondary current (*I_ts_*) was converted to voltage difference on the resistances *R*_0_. Because of the properties of the almost infinite open loop gain of the op-amp *A*_1_ and the used feed-back by *R_f_*, it follows that:(19)$$V_{2}-V_{1}=2R_{0}I_{ts}$$
where *R*_0_ indicates the resistances in the input loop of the operational amplifier *A*_1_ and *I_ts_* = *I_t_*/300 ($V_{2}-V_{1}=2R_{0}I_{ts}\Rightarrow V_{2}-V_{1}=0.47\;\mathrm{mV}$). Additionally, the alternative input signal must be transformed into a unipolar signal appropriate for A–D conversion of voltage at the output of the measurement system (*U_oA_*_3_). Figure 9 shows the scheme of the chosen electronic circuit. The frequency analyses of the electronic circuit were considered in order to design a suitable gain and band-pass. Assuming that the operational amplifiers were the same type, and after analyses of the scheme in Figure 9, it follows:(20)$$U_{o}=\frac{A_{\beta }\left(\beta_{1}s+1\right)}{\alpha_{2}s^{2}+\alpha_{1}s+1}\left(V_{2}-V_{1}\right)+\frac{U_{p}}{2}$$
where *A*_β_ = *R_f_*/*R*_0_ represents closed-loop gain, β_1_ = 1/ω*_t_*, α_2_ = 1/(*B*ω*_t_*^2^), α_1_ = 1/(*B*ω*_t_*), $B=R_{0}/(R_{F}+R_{0})$ ($R_{0}=47\Omega ,\;R_{F}=100\;k\Omega$), ω*_t_* is unity-gain bandwidth ($\omega_{t}=2\pi f\;\Rightarrow$
$\omega_{t}=2\pi 4\times 10^{5}\;rad/s$ (from the data sheet)) (usually specified on the data sheets of op. amps.), and *U_p_* denotes the power supply voltage. Using (20) and considering that s→0, the output voltages can now be calculated as:(21)$$U_{o}=\frac{R_{f}}{R_{0}}\left(V_{2}-V_{1}\right)+\frac{U_{p}}{2}$$

Based on (21) the output voltage range of the chosen instrumental amplifier gain could be defined for the whole range of the leakage current ($U_{0}=\frac{R_{f}}{R_{0}}\left(V_{2}-V_{1}\right)+U_{p}/2\overset{}{\Rightarrow }U_{0}\in \left(0.5V,\;2.5V\right)\;when\;I_{t}\in \left(-3mA,\;3mA\right)$). Additionally, based on (20), the transfer function for alternative difference signal was considered when DC offset (*U_p_*/2 = 0) was excluded, as follows:(22)$$H_{0}\left(s\right)=\frac{U_{o}(s)}{V_{2}(s)-V_{1}(s)}=\frac{A_{\beta }\left(\beta_{1}s+1\right)}{\alpha_{2}s^{2}+\alpha_{1}s+1}\Rightarrow H_{0}\left(s\right)\approx \frac{A_{\beta }}{1+\frac{s}{\omega_{b}}}$$
where ω*_b_* represents the small-signal bandwidth frequency (*f_b_* = 188 Hz), which can be obtained after the short calculation:(23)$$\omega_{b}=\frac{\omega_{t}}{\left(1+\frac{R_{f}}{R_{0}}\right)}\Rightarrow f_{b}=\frac{f_{t}}{\left(1+\frac{R_{f}}{R_{0}}\right)}$$

For certain data parameters and using MATLAB, the frequency characteristics were calculated, and they are shown in Figure 10.

Unfortunately, the op-amp devices consist of non-ideal components that can introduce some deviation into the measurement results. The sensitivities function $S_{x}^{y}$ can be used to quantify this inconsistency. This function is defined as:(24)$$S_{x}^{y}=\frac{\partial y}{\partial x}\frac{x}{y},$$
where *x* denotes the variable of interest. According to (22), a closed-loop gain *A*_β_, a frequency ω*_b_*, and *y* denote the circuit parameters, as is the absolute form of the function H(jω), as follows:(25)$$H_{\beta }=\left|H(j\omega )\right|=\frac{A_{\beta }}{\sqrt{1+{\left(\frac{\omega }{\omega_{b}}\right)}^{2}}}$$

Using (24) and according to the variables of interest, two sensitivities ($S_{A_{\beta }}^{H_{\beta }}$ and $S_{\omega_{b}}^{H_{\beta }}$) need to be evaluated:(26)$$S_{A_{\beta }}^{H_{\beta }}=\frac{\partial H_{\beta }}{\partial A_{\beta }}\frac{A_{\beta }}{H_{\beta }}=1$$
(27)$$\begin{array}{ll} S_{\omega_{b}}^{H_{\beta }}=\frac{\partial H_{\beta }}{\partial \omega_{b}}\frac{\omega_{b}}{H_{\beta }}=\frac{\omega^{2}}{\omega_{b}^{2}+\omega^{2}}; & \;\mathrm{when}\;\omega =0\Rightarrow S_{\omega_{b}}^{H_{\beta }}=0;\\ & \;\mathrm{when}\;\omega =\omega_{b}\Rightarrow S_{\omega_{b}}^{H_{\beta }}=\frac{1}{2} \end{array}$$

From (26), the sensitivity of the low-frequency gain change can be evaluated, so the *A*_β_ depends on the resistances *R_f_* and *R*_0_; if these change by ±1%, the *A*_β_ changes by ±2, so it follows
(28)$$\frac{\partial H_{\beta }}{H_{\beta }}=S_{A_{\beta }}^{H_{\beta }}\frac{\partial A_{\beta }}{A_{\beta }}=\pm 2\%$$

Based on (27), the other sensitivities‘ effects can be evaluated for two cases;
(29)$${\left.\frac{\partial H_{\beta }}{H_{\beta }}\right|}_{\omega =0}=S_{A_{\beta }}^{H_{\beta }}\frac{\partial \omega_{b}}{\omega_{b}}=0$$
(30)$${\left.\frac{\partial H_{\beta }}{H_{\beta }}\right|}_{\omega =\omega_{b}}=S_{A_{\beta }}^{H_{\beta }}\frac{\partial \omega_{b}}{\omega_{b}}=-\frac{1}{2}\frac{\partial \omega_{b}}{\omega_{b}}\approx 4.7\times 10^{-6}$$

Accordingly, the worst-case precision of the whole analogue part of the SAMD is evaluated in (28), and it only depends on the precision of the closed-loop gain *A*_β_ and, consequently, on the resistances *R_f_* and *R*_0_.

#### 3.1.3. Microcontroller and Communication Unit

A digital signal acquisition device contains a microcontroller (Ultra-low-power microcontroller STM32L433C, STMicroelectronics, Geneva, Switzerland) and communication module (Ultra-small LCC Quad-band Module QUECTEL MC60, Quectel Wireless, Shanghai 200233, China). The ultra-low-power microcontroller is equipped with:
12-bit analogue-digital converter (ADC),operational amplifier,two ultra-low-power comparators with reach communication interfaces, anddevelopment supports, etc.


The microcontroller unit is also capable of operating in a temperature range of −40 to +85 °C. The microcontroller could also communicate with the periphery by using TTL gates. All of the terminals should be protected against over-voltages [29].

The communication unit is based on a GSM/GPRS/GNSS module, connected with the microcontroller over three embedded synchronous receivers/transmitters that were incorporated in the microcontroller chip. These transmitters/receivers enabled the use of asynchronous communication protocol. The GSM module can also use a set of Internet protocols. The main feature of this module is low power consumption, which was the main issue for using it in the Surge Arrester Monitoring Devices [34].

#### 3.1.4. Battery

A lithium-thionyl chloride battery was used, with a nominal voltage of 3.6 V and nominal capacity of 6 Ah. The battery (Saft LS17500 A 3.6V Primary Lithium Battery) that was used was distinguished by its greater energy density and operating temperature range between −60 °C and +150 °C. The battery producer claims that, with the current system running at a temperature of −40 °C, the battery would last at least 10 years, if not more (two in parallel enable c.c.a. 20 years of autonomy). The battery is not intended to be replaced during the SAMD’s lifetime [31]. Further development is under way to replace the mentioned battery with an energy harvesting system that will, theoretically, prolong the lifespan of the device indefinitely.

## 4. Experimental Results

For verification of the described measuring principle that is based on (15) to (18), the experimental test-bench system was built, as shown in Figure 11. The Surge Arrester Monitoring Device prepared the current measurement results into the form that is required for further processing. The measurement results were processed with the acquisition units and microcontroller, as described in Section 3. Subsequently, the results were forwarded to the computer in the “distribution” centre, where they were recalculated to achieve the right values of the resistive current component.

### 4.1. Experimental Test-Bench

The proposed algorithm was verified using the test-bench system. For verification purposes, the arrester was measured using the classical oscilloscope measurement (as a reference system), and it was also measured by using the proposed described system for extraction of the resistive current component ${\hat{I}}_{r}$. The surge arrester was connected, as indicated in Figure 2 for reference measurement (oscilloscope), and as indicated in Figure 7 for the surge arrester monitoring device. Both of the measurements were performed at the same operating points.

### 4.2. Calibration of the Measurement Devices

The signals are provided via the analogue acquisition system to the ADC of the microcontroller. The used ADC allows for a voltage between 0 and 3 V on the input terminals. Because of this, a single power-supply voltage was also used for safety reasons for the analogue acquisition circuit. As a consequence of this, the offset voltage of *U_p_*/2 was added to the analogue signal, which represents the leakage current (*I_t_*) information. After the ADC conversion, the obtained measurement results were sent using the communication unit to the “distribution centre” (computer or server). The information that was sent to the server was coded as a series of numbers representing the leakage current I_t,_ but in the unit of quants not in the μA. Definitely, this information must be calibrated in the proper unit μA [34]. For calibration purposes, the measurement constants were calculated from the properties of the entire measurement chain that are described in Section 3.1.

#### 4.2.1. Measurement Constant

Figure 12 shows the block scheme of the calibration process. The analogue signal was prepared for ADC conversion in the signal acquisition circuit. The alternative current signal was amplified and shifted by the offset voltage, as follows:(31)$$I_{t}\in \left(I_{t,\min}\;I_{t,\max}\right)\Rightarrow \left(V_{2}-V_{1}\right)=U_{d}\in \left(U_{I}{}_{t,\min}\;U_{I}{}_{t,\max}\right)$$
(32)$$U_{0}=\frac{R_{f}}{R_{0}}U_{d}+Up/2\Rightarrow U_{0}\in \;\left(U_{0}{}_{,\min}\;U_{0,\max}\right)$$

Afterwards, the obtained result was converted to a digital signal using the 12-bit ADC. The offset voltage of 1.5 V corresponded with:(33)$$U_{0,offset}=U_{p}/2\Rightarrow 2500=0000\;1001{\;1100\;0100}_{binary}$$
when the *U*_0,min_ (0.5 V) was applied on the ADC, the input was obtained for:(34)$$U_{0,\min}=0.5\;V\Rightarrow 452=00{00\;0011\;1000\;0100}_{binary}$$
and when the *U*_0,max_ (2.5 V) was applied on the ADC, input was obtained for:(35)$$U_{0,\max}=2.5\;V\Rightarrow 4548=000{1\;0001\;1100\;0100}_{binary}$$

The upper voltage was limited to 4096 due to the 12-bit quantisation, which means up to the value when the output voltage of the analogue acquisition circuit reached 2.13 V. Carefully following the above-described digitalisation from (24) to (28), it can be concluded that the obtained results, after removing the offset values, should be multiplied by the constant (3000/2048) in order to reconstruct the measured leakage current. Hence, according to this, and also considering the variables indicated in Figure 12, it follows that;
(36)$$Y=X-2500\Rightarrow I_{t}\left[\mu A\right]=\frac{I_{t,\max}\left[\mu A\right]}{4548-2500}Y=\frac{3000}{2048}Y=1.464844\;Y.$$

Because of the above mentioned 12-bit quantisation, the input current range should be corrected to:(37)$$I_{t}\in \left(I_{t,\min L}I_{t,\max L}\right)=\left(-2300\;\mu A\;2300\;\mu A\right)$$

In practice, the surge arresters are 100% out of order when *I_t_* reaches a magnitude over 1500 μA, so this range guarantees the appropriate consideration of the surge arresters’ “health” while measuring the resistive component of the leakage current.

#### 4.2.2. Verification of the Calibration Process

The obtained results should be verified in order to justify the measurement principle. Three different operating points of the surge arresters were measured with the oscilloscope and with the Surge Arrester Monitoring Device. Figure 13a–c show the results of both measurement principles. The current and voltages’ waveforms of the surge arrester measured by oscilloscope are shown in the first row. The results from the oscilloscope CSV files are treated as reference, and they are slightly different from the results obtained from the SAMD, as can be noticed from Figure 13a–c. The reason is the pass-band of the used analogue signal acquisition circuit. After inspection of the performance index shown in Table 5, it was concluded that the obtained results reflected the measured currents accurately when comparing them with the scope reference values.

Greater deviation appeared when the leakage current had a higher magnitude that corresponded with the magnitudes of resistive components of ${\hat{I}}_{r}$, which was much greater than the 600 μA. The last row oscilloscope measured value *I_t_avg_* corresponds with the resistive component of ${\hat{I}}_{r}$ = 783 μA.

### 4.3. Measurement Results

The whole Surge Arrester Monitoring Device was tested under laboratory conditions. In order to verify the measurement results, these were compared with the reference resistive current component ${\hat{I}}_{r}$ that was measured by an oscilloscope. The leakage current was calibrated appropriately, and the procedure described in Section 2.3 was applied from the files that were obtained from the SAMD. The resistive component of leakage current was calculated using (15) to (18). Table 6 summarises the measurement results. The performance indexes (absolute and relative errors) were calculated, as follows:(38)$$\varepsilon_{abs}={\hat{I}}_{r\_cal}-{\hat{I}}_{r}$$
and
(39)$$\varepsilon_{rel}=\frac{{\hat{I}}_{r\_cal}-{\hat{I}}_{r}}{{\hat{I}}_{r}}\times 100$$

The obtained results had an expected precision, and the measurement principle with the proposed electronic devices can be used for on-line measurement of the resistive component of the surge arrester leakage current.

## 5. Discussion

The measurement principle that is presented here enables the measurement of the resistive component of the MOSA leakage current successfully without the use of voltage measurement as a reference, as shown in Table 4. The error of the resistive component of leakage current fell well within the Standard Deviation range of $\left(-2\sigma_{\hat{I}r},2\sigma_{\hat{I}r}\right)$. The errors had a tendency to decrease as the resistive component of the total leakage current increased, as also seen in Table 4. This is a welcome outcome, when considering that, at lower values of the resistive current, the significance of the state of the MOSA does not play a vital role in the maintenance procedure. However, as the resistive component of the total leakage current increases, it is imperative to determine the state of the MOSA accurately. Transferring data from the SAMD to the server requires the use of a database system. The sampling rate of leakage current is hourly. For a single arrester, the distributor will be faced with enormous amounts of information, which must be processed systematically in order to obtain the adequate values from the measurement system. This would allow for the user to take full advantage of the smart system and be able to monitor the MOSA in even greater detail [32,33,34,35].

## 6. Conclusions

The MOSA itself is a relatively inexpensive part of the power grid, but its hindered performance can lead to increasingly greater costs. Maintenance plays a very important role in the distribution and transmission networks, as mentioned before. With the correct insight into the state (health) of the power grid, adequate measures can assure the continuous working of the grid without power outages and high cost maintenance interventions, thus reducing the overall costs of the power grid. The goal was to develop a mechanism to extract the resistive component of the leakage current of a MOSA without the use of voltage as a reference. Usually, the MOSA leakage current and its resistive component measurement is only performed in a laboratory environment, because the method also needs voltage measurement. For this method, the static characteristic, obtained as a result of the analyses performed in Section 2, is required to be measured by the manufacturer of the MOSA. Alternatively, the static characteristic can be established by extensive testing of the MOSA in question, as described in Section 2.3 of the article. The analogue part of the electronic devices is designed with a high gain, because the leakage current represents a small quantity as compared to the current flowing through the arrester when it is operating due to overvoltage (a range of 1000 µA compared with a few 10 kA). Using the described principle, it is possible to design a MOSA monitoring device and equip every surge arrester that is mounted in the distribution and transmission grid as an integrated part of the arrester. The prototype housings have been 3D printed and tested in a high-voltage laboratory, achieving an IP level of 67. The prototype has also been tested in a real-life working environment, where it was exposed to both different electrical (atmospheric discharges, switch devices’ manipulations, etc.) as well as environmental (moisture, dust, temperature, etc.) elements. The obtained results were promising, and they are an excellent guide for design of a reliable, precise, and cheap SAMD, appropriate for every installed surge arrester in the high-voltage transmission and distribution grid.

## Figures and Tables

**Figure 1 sensors-21-01257-f001:**
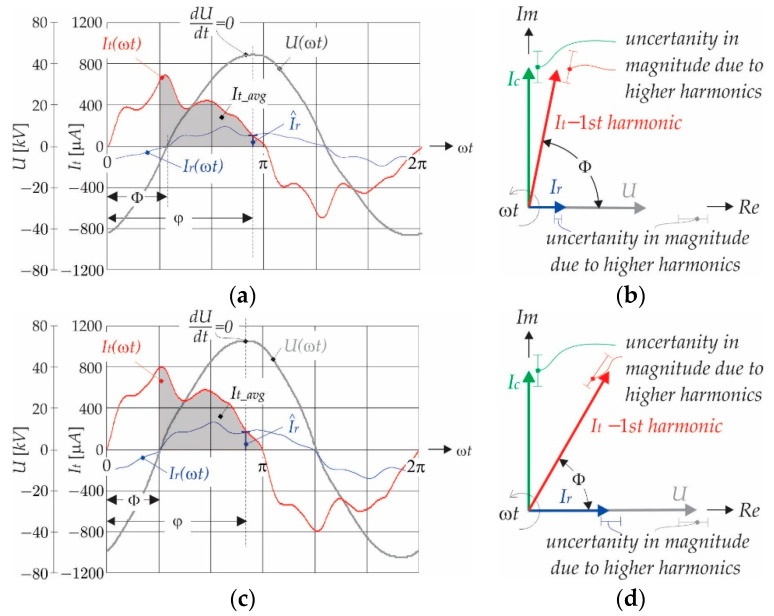
(**a**) Current and voltage wave shape when *U* = 27 [kV]_rms_; (**b**) Phasor diagram, when Φ = 72°; (**c**) Current and voltage wave shape when *U* = 37 [kV]_rms_; and, (**d**) Phasor diagram, when Φ = 64°.

**Figure 2 sensors-21-01257-f002:**
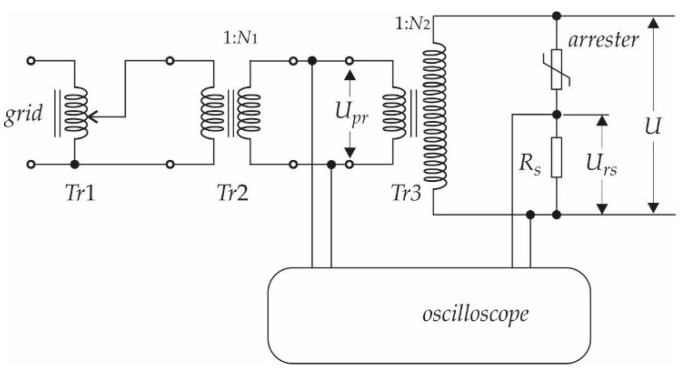
Measurement set-up scheme.

**Figure 3 sensors-21-01257-f003:**
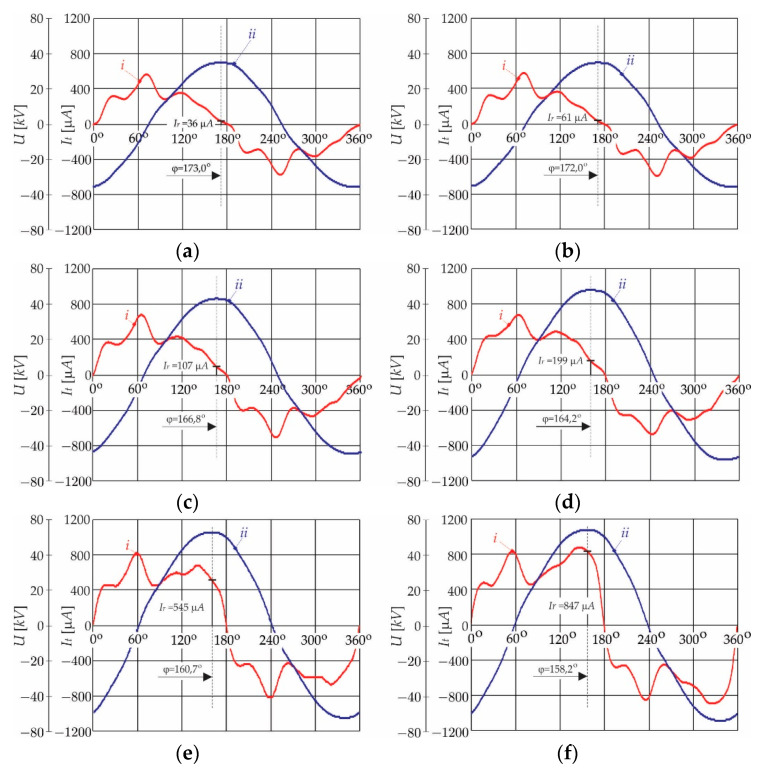
Measured current (*I_t_* ) and voltage (*U*) with oscilloscope (CSV files) *I_t_*: ***i*** arrester leakage current; ***ii*** measured arrester voltage (*U*): (**a**) at *U* = 25 kV_rms_, φ = 173.0°, ${\hat{I}}_{r}$ = 61 μA; (**b**) at *U* = 27 V_rms_, φ = 172.0°, ${\hat{I}}_{r}$ = 61 μA; (**c**) at *U* = 31 kV_rms_, φ = 166.8°, ${\hat{I}}_{r}$ = 107 μA; (**d**) at *U* = 36 kV_rms_, φ = 164.2°, ${\hat{I}}_{r}$ = 199 μA; (**e**) at *U* = 37 kV_rms_, φ = 160.7°, ${\hat{I}}_{r}$ = 545 μA; (**f**) at *U* = 39 kV_rms_, φ = 158.2°, ${\hat{I}}_{r}$ = 847 μA.

**Figure 4 sensors-21-01257-f004:**
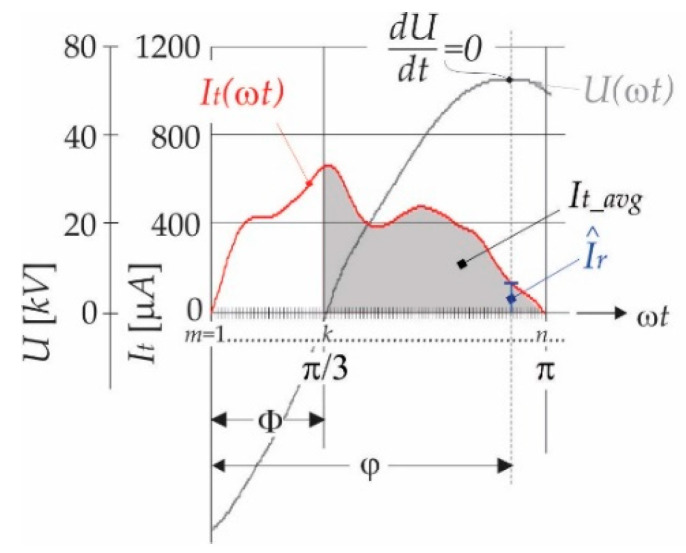
Description of average value *I_t_avg_* and its relation with pair (φ, ${\hat{I}}_{r}$).

**Figure 5 sensors-21-01257-f005:**
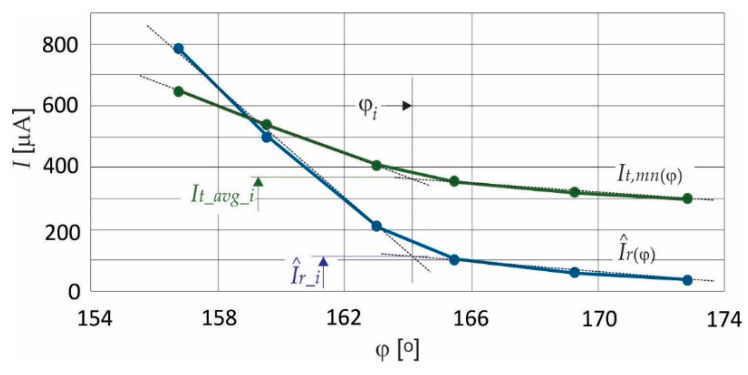
Static characteristics of ${\hat{I}}_{r}={\hat{I}}_{r}\left(\varphi \right)$ and $I_{t,mn}=I_{t,mn}\left(\varphi \right)$.

**Figure 6 sensors-21-01257-f006:**
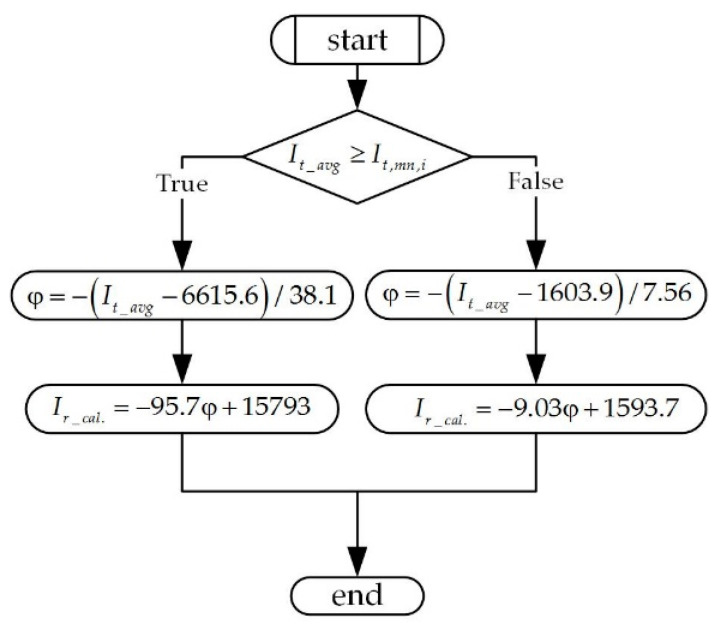
UML diagram for the functional mapping algorithm, $I_{t\_avg}\to \varphi \to {\hat{I}}_{r\_cal.}$.

**Figure 7 sensors-21-01257-f007:**
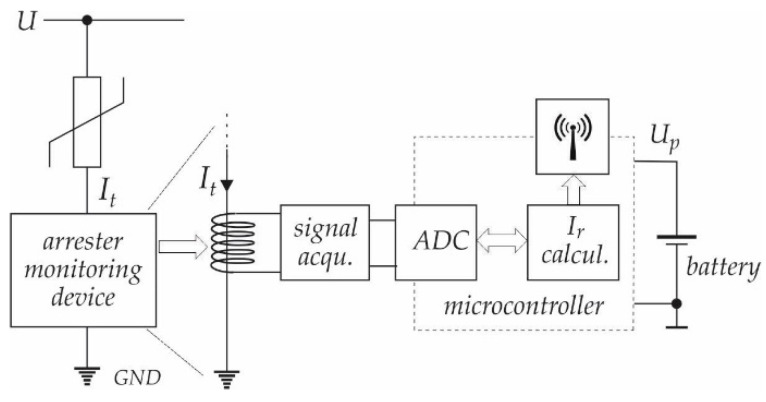
The block scheme of Surge Arrester Monitoring Device (SAMD).

**Figure 8 sensors-21-01257-f008:**
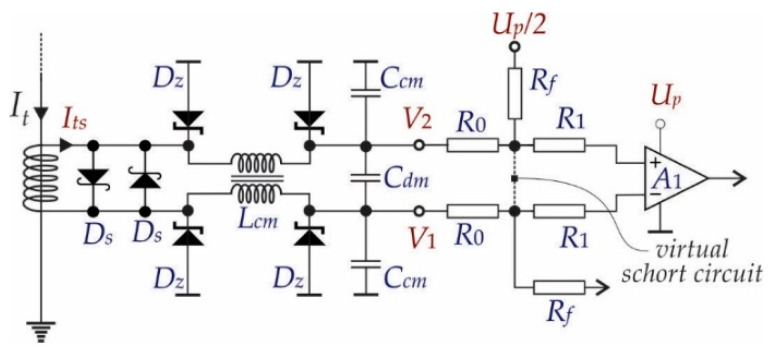
Protection and Common Mode (CM) and Differential Mode (DM) noise reduction circuit.

**Figure 9 sensors-21-01257-f009:**
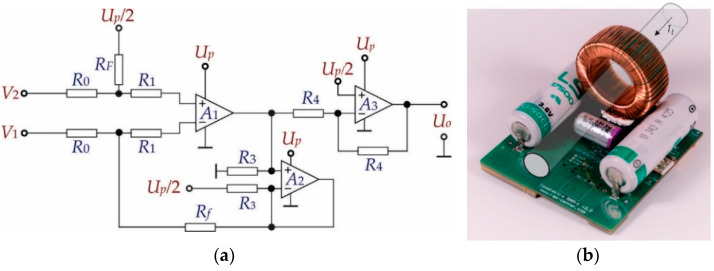
(**a**) Analogue signal acquisition circuit; (**b**) PCB (printed circuit board) with components of the measurement device.

**Figure 10 sensors-21-01257-f010:**
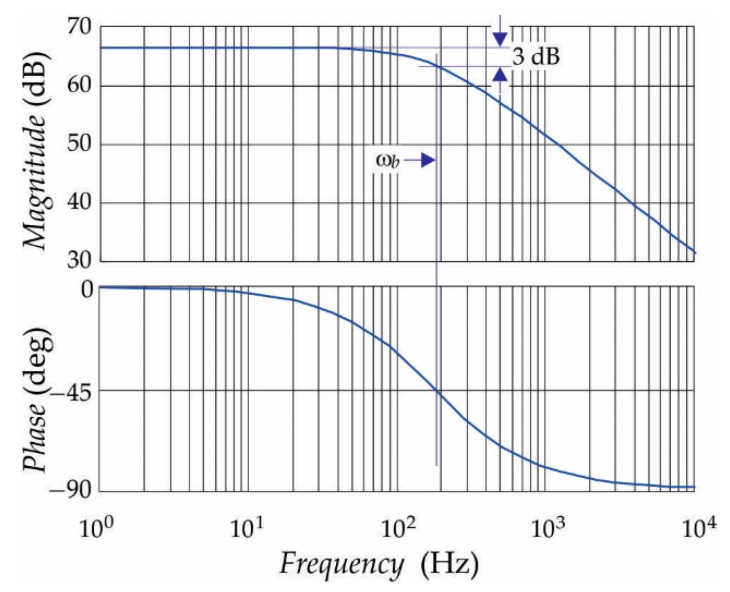
Bode diagram of *H_o_*(s).

**Figure 11 sensors-21-01257-f011:**
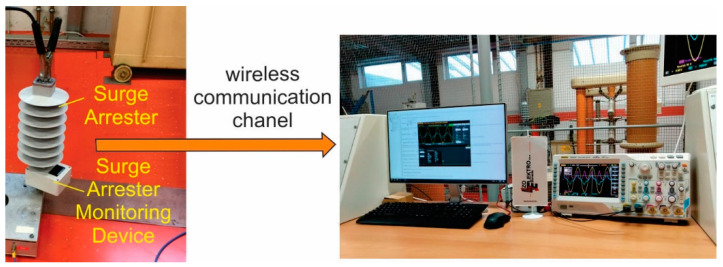
Measurement set-bench.

**Figure 12 sensors-21-01257-f012:**
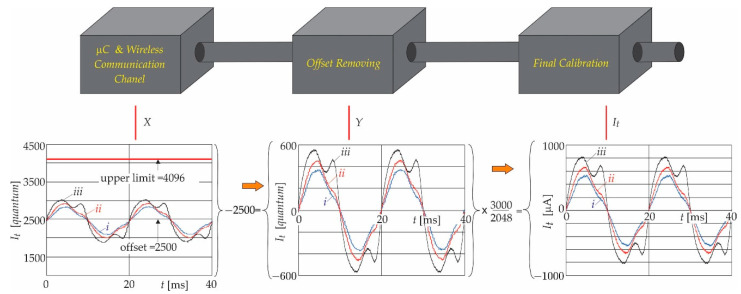
Calibration procedure; the roman letters *i*, *ii* and *iii* indicate measurements at 0.80*U*, 1.00*U*, and 1.20*U*, respectively.

**Figure 13 sensors-21-01257-f013:**
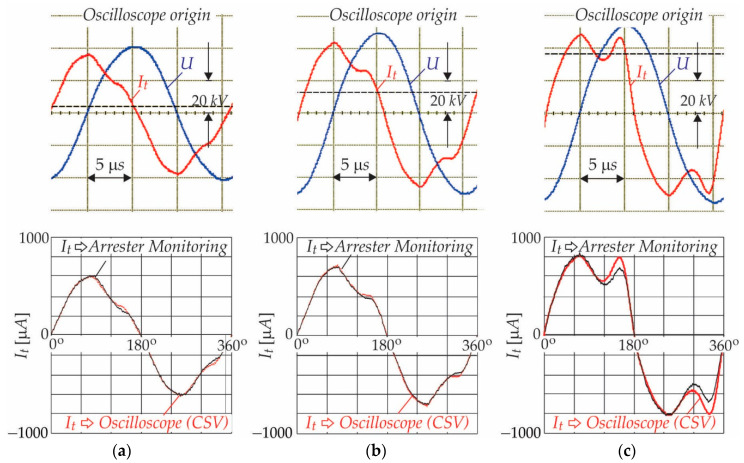
Verification of the calibration process when the supply voltages to the surge arrester were: (**a**) 0.9 *U*; (**b**) 1.1 *U*; and, (**c**) 1.275 *U*.

**Table 1 sensors-21-01257-t001:** Annual losses due to faulty surge arresters in the Slovenian distribution network.

	Annual for 1 MOSA	Annual for 12,090 pcs of MOSA
Power losses [MWh]	0.486	5.872
Financial losses [€]	35.4	428.312
Carbon footprint [t]	0.133	16.4416

**Table 2 sensors-21-01257-t002:** Resume of measured phase delay φ and resistive component ${\hat{I}}_{r}$ extracted from the oscilloscope.

Sample 1		Sample 2		Sample 3		Sample 4		…	Sample N		Mean Val.			Relative σ	
φ_1_ [°]	${\hat{I}}_{r,1}$ [μA]	φ_2_ [°]	${\hat{I}}_{r,2}$ [μA]	φ_3_ [°]	${\hat{I}}_{r,3}$ [μA]	φ_4_ [°]	${\hat{I}}_{r,4}$ [μA]	.…	φ_N_ [°]	${\hat{I}}_{r,N}$ [μA]	φ [°]	${\hat{I}}_{r}$ [μA]	*U*_rms_ [kV]	σ_φ_ [%]	$\sigma {\hat{I}}_{r}$ [%]
157	812	158	847	158	847	155	755	….	156	750	157	802	39	0.31	3.9
159	510	160	521	161	545	160	475	….	159	480	160	500	37	0.36	4.3
162	220	163	214	165	199	162	195	….	163	183	163	202	36	0.59	7.3
165	107	166	102	167	107	164	95	….	166	102	166	103	33	0.67	4.8
168	66	169	56	169	61	171	56	….	171	60	170	59	31	0.88	5.8
173	33	172	33	173	36	174	36	….	173	37	173	35	25	0.39	5.5

**Table 3 sensors-21-01257-t003:** Resume of the calculated average values of leakage current *I_t_*, measured with an oscilloscope in the intervals $60^{{}^{\circ}}\leq \varphi \leq 180^{{}^{\circ}}\;(\pi /3\leq \omega t\leq \pi )$, according to the relation $I_{t\_avg}=I_{t\_avg}\left(\varphi \right)$.

Sample 1		Sample 2		Sample 3		Sample 4		…	Sample N		Mean			Relative σ	
φ [°]	*I*_*t_avg*,1_ [μA]	φ [°]	*I*_*t_avg*,2_ [A]	φ [°]	*I*_*t_avg*,3_ [μA]	φ [°]	*I*_*t_avg*,4_ [μA]	…	φ [°]	*I*_*t_avg,N*_ [μA]	φ [°]	*I_t,mn_* [μA]	*U*_rms_ [kV]	σ_φ_ [%]	**σ***_It,mn_* [%]
157	643	158	665	158	664	155	621	…	156	640	157	646	39	0.73	2.5
159	546	160	548	161	545	160	523	…	159	522	160	537	37	0.43	2.2
162	426	163	420	165	418	162	380	…	163	401	163	409	35	0.68	4.1
165	359	166	353	167	356	164	340	…	166	365	166	355	33	0.46	2.4
168	329	169	319	169	318	171	314	…	171	321	170	320	31	0.72	1.5
173	298	172	294	173	301	174	305	…	173	310	173	299	25	0.27	2.0

**Table 4 sensors-21-01257-t004:** Verification of measurement results.

Measured Values	Phase	Calculated (15)–(18)	Scope Value	Relative Error
*I_t_avg_* [μA]	φ [°]	${\hat{I}}_{r\_cal.}$ [μA]	${\hat{I}}_{r}$ [μA]	ε [%]
665	156.25	839	847	0.9
643	156.86	783	812	3.6
522	160.01	480	480	0.0
418	162.74	218	199	−9.8
356	165.06	103	107	3.5
319	169.95	59	56	−5.4
298	172.72	34	33	−3.0
570	158.75	600		

**Table 5 sensors-21-01257-t005:** Experimental results’ verification.

SAMD Measured	Oscilloscope Measured	Relative Error
*I_t_avg_* [μA]	*I_t_avg_* [μA]	ε_rel_ [%]
327.3	331.4	−1.24
372.1	375.8	−0.98
605.2	642.5	−5.81

**Table 6 sensors-21-01257-t006:** Experimental results’ verification.

SAMD Measured	Calculated (15)–(18)	Reference Scope Value	Absolute $\mathrm{Error}\;\mathrm{of}\;{\hat{I}}_{r}$	Relative $\mathrm{Error}\;\mathrm{of}\;{\hat{I}}_{r}$
*I_t_avg_* [μA]	${\hat{I}}_{r\_cal.}$ [μA]	${\hat{I}}_{r}$ [μA]	ε_abs_ [μA]	ε_rel_ [%]
327.3	69.0	79.1	10.1	−12.8
372.1	103.2	97.0	6.2	6.3
409.7	197.0	195.3	1.7	0.9
411.1	201.1	204.2	−3.1	−1.6
457.0	316.5	311.0	5.5	1.6
581.2	628.5	628.0	0.5	0.1

## Data Availability

Data is contained within the article at hand.

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
