# Peer review of "Extracting the Resistive Current Component from a Surge Arrester’s Leakage Current without Voltage Reference"

_sensors, 2021, doi:10.3390/s21041257_

Round 1

Reviewer 1 Report

Article is about new technique to measure resistive current component of surage arrester

The article is well formatted and properly structured. State of the art is well done. The idea of the proposed measurement is presented correctly and justified. The tests are performed correctly. The discussion of the results is correct.

The following points can be clarified better:

(line 238) Why is the current transformer ratio 1: 300 ?

(line 302 - 307) the authors write that they use a GSM modem. While earlier and afterwards, they inform that the data to the server is sent via WiFi

What is the sampling frequency and how often the data is sent to the server?

(line 349) What is the reference voltage of the ADC

END

Reviewer 2 Report

The topic of this paper is interesting to the readers, within the scope of the journal, however prior its publication many changes must be done.

The use of English must be improved. The paper includes some grammatical and syntax errors.

The Introduction must be revised. The authors must present the general research area to unfamiliar readers and at most to present the current state-of-the-art in order to show the contribution/novelty of their work. Authors must omit old reference (more than 15 years old), describe/analyse more the current mentioned references and include many more related references, such as the following:

Vita V., Christodoulou C.A., Comparison of ANN and finite element analysis simulation software for the calculation of the electric field around metal oxide surge arresters, Electric Power Systems Research, Vol. 133, 2016, pp. 87-92.

The discussion section must comment more the produced results.

A Conclusions section must be included summarizing the work presented within the paper.

Reviewer 3 Report

This article addresses remote monitoring of MOSA (Metal Oxide Surge Arrester). It is well structured and written, has no significant style errors, overall is fine. However, there are several issues from scientific perspective and research design method, as identified below.

1. MOSA represents a necessary part of the electric power grid. From this perspective, this work might be of interest for engineers and researchers. However, studying the IEEE literature which is the most appropriate in this regard, we can notice that there are very few articles focused on MOSA published during the last 2 decades and especially over the last 5-7 years. In addition, most of them focus on MOSA modeling and, interesting enough, they lack citations. There are many articles having 0 citations, including journal articles. Such situation raises pertinent questions about the relevance of this topic and the possible impact of this work.

2. As mentioned above, there are more recent articles indexed to IEEE platform, including 2020, hence the list of references is not appropriate since it reports older references. Actually, the first 14 references are too old while the newest reference dates back in '15.

3. The authors state that "it is possible to claim that the measurement principle leads to a so-called “smart” sensor". In fact, it is obvious the effort of implementing a smart solution for MOSA, yet not complete (!). However, this work dealing with smart things is not supported by scientific references, i.e. articles or books, focused on smart city and smartification, especially proving the Industry interest for power grid smartification (eventually EU regulations and standards). For instance, I would suggest the following references:

[1] "Data acquisition and signal processing for smart sensors" (Wiley, 2002)

-> Fig. 1.2 clearly shows the structure of a smart sensor, it contains the sensing element, signal conditioning and uController, it is quite important to clarify what we name "smart sensor", it is not clear in this work

[2] Smart Sensors Networks (Wiley, 2017) 

-> this perspective is important because any smartification step raises further questions about how the smart sensor do communicate with monitoring system and eventually to each other, what protocol should be adopted/developed in case that dozens of sensors are connected together, cybersecurity etc

[3] Smart Grids – Fundamentals and Technologies in Electricity Networks (Springer, 2014)

-> this reference mentions MOSA in a single dedicated section, i.e. pp. 64-70

-> interestingly, this book mentions "smart metering", "smart distribution", etc.

[4] Smart sensors for Industrial Applications (CRC Press, 2013)

-> it includes, among other technical details and hints, the calibration procedure for some sensors, from this perspective it is not clear how calibration would be conducted for the "smart sensor" proposed in this article

4. Taking into account that this topic is highly practical and very low citations score on IEEE platform, I have searched for MOSA in a well known database of patents, i.e. freepatentsonline.com. I have to say that the database returned more than 500 patents and patent applications containing "surge arrester" and "monitoring" in their title. The first and most interesting solution is disclosed by "Surge arrester condition monitoring", available at:

Interestingly, this patent application filed in 2009 discloses a similar architecture of wireless monitoring of MOSA, as many other patents do, including also a temperature sensor and hygrometer. In other words, a patent application already filed many years ago discloses a solution covering the architecture proposed in this article and even offering extra benefits. A screenshot taken from that patent application is herein enclosed. When comparing scientific articles with patent applications, it does not matter how the monitoring sensor is to be connected to MOSA, it seems different, but the overal architecture disclosed and protected by patent application.

5. The solution proposed in this article reports the case of a single MOSA monitoring. It is not clear how the overal monitoring scheme will look like when instantiating multiple MOSAs. For instance, in a real implementation, we are supposed to use many surge arresters. I attached a picture with a power plant available in my area. What is the monitoring plan in case of such power plant with dozens of MOSA? What is the cost associated to integrating dedicated sensors for monitoring all these surge arresters?

6. The circuit used to monitor the sensor is biased by means of a battery. This raises 2 possible questions:

6.1 What battery was used for experiment? If taking into account the previous remark, including multiple batteries to cover a power plant would increase the maintenance cost. In addition, the battery model reported by authors is more expensive. In the long term, this will raise issues about the implementation cost.

6.2 Fig. 11 and all reported results address an indoor experimental scenario, this means a perfect environment for humans, i.e. moderate temperature (25-28 degrees) and humidity. The problem here is that such monitoring device is supposed to work in outdoor environment, attached to MOSA, as disclosed in the patent application as well. This means, depending on countries, a large spread in terms of temperature and humidity, therefore generating extra measurement errors. How about the aging process? 

From all these perspectives, I do believe that this experimental scenario is limitative, the implementation is not new compared with solutions disclosed by patents and the research procedure is not well conducted from "smart" concept perspective (not being supported by relevant references on smart city). In addition, this sector is ruled by important business players, such as Siemens, this company still delivers reliable solutions for power grid. It is not easy to compete with them.

Round 2

Reviewer 2 Report

The authors have conducted the requested changes.

The paper has been significantly improved.

It can be accepted for publication.

Reviewer 3 Report

The article has been updated in accordance with my previous comments so I think it could be published in this form.